# The Epithelial Cell Leak Pathway

**DOI:** 10.3390/ijms22147677

**Published:** 2021-07-18

**Authors:** Ashley Monaco, Ben Ovryn, Josephine Axis, Kurt Amsler

**Affiliations:** 1Department of Biomedical Sciences, New York Institute of Technology College of Osteopathic Medicine, Northern Boulevard, Old Westbury, NY 11568, USA; amonaco@nyit.edu (A.M.); jaxis@nyit.edu (J.A.); 2Department of Physics, New York Institute of Technology, Northern Boulevard, Old Westbury, NY 11568, USA; bovryn@nyit.edu

**Keywords:** paracellular permeability, tight junction, Pore Pathway, Leak Pathway, occludin, ZO-1, ZO-2, claudin

## Abstract

The epithelial cell tight junction structure is the site of the transepithelial movement of solutes and water between epithelial cells (paracellular permeability). Paracellular permeability can be divided into two distinct pathways, the Pore Pathway mediating the movement of small ions and solutes and the Leak Pathway mediating the movement of large solutes. Claudin proteins form the basic paracellular permeability barrier and mediate the movement of small ions and solutes via the Pore Pathway. The Leak Pathway remains less understood. Several proteins have been implicated in mediating the Leak Pathway, including occludin, ZO proteins, tricellulin, and actin filaments, but the proteins comprising the Leak Pathway remain unresolved. Many aspects of the Leak Pathway, such as its molecular mechanism, its properties, and its regulation, remain controversial. In this review, we provide a historical background to the evolution of the Leak Pathway concept from the initial examinations of paracellular permeability. We then discuss current information about the properties of the Leak Pathway and present current theories for the Leak Pathway. Finally, we discuss some recent research suggesting a possible molecular basis for the Leak Pathway.

## 1. The Tight Junction Is the Paracellular Permeability Barrier

The concept of a barrier, the “terminal bar”, that restricts the movement of solutes and water through the paracellular pathway between epithelial cells dates back at least to Bizzozero [1]. Identification of the tight junction (*zonula occludens)* as the most apical portion of the terminal bar which restricts the paracellular movement of large molecules between body compartments separated by epithelial cell sheets was proposed by Farquhar and Palade [2]. Their pioneering experiments demonstrated by electron microscopy that hemoglobin and zymogen could not penetrate the lateral intercellular spaces beyond the tight junction structure in several epithelial tissues. They stated, “Hence the tight junction is impervious to concentrated protein solutions and appears to function as a diffusion barrier or ‘seal’”. Ussing and Windhager [3] demonstrated that chloride ions moved across the frog skin epithelium via the paracellular pathway to maintain charge neutrality following active sodium ion transport. This showed that the tight junction exhibited permeability to some solutes. Machen et al. [4] used lanthanum ions (La^+3^) to examine the permeability of the tight junction in several epithelia. La^+3^ ions are not transported into cells and can be precipitated into the electron dense lanthanum sulfate which can be detected by electron microscopy [5]. They demonstrated that La^+3^ permeates the tight junctions of both intestinal and gallbladder epithelia, supporting the idea that the tight junction is the site of the epithelial tissue paracellular permeability pathway. It was previously shown that lanthanum ions did not permeate into the tight junctions of toad urinary bladder epithelia [4] and frog skin epithelia [6]. Based on these and other observations, Machen et al. [4] and Frömter and Diamond [7] proposed a categorization of epithelia as “leaky” or “tight” based on their relative permeability to small ions. “Leaky” epithelia exhibit a substantially greater paracellular small ion permeability as compared to the transcellular small ion permeability. They exhibit a relatively low transepithelial resistance. In contrast, “tight” epithelia exhibit an equivalent or even greater transcellular small ion permeability as compared to the paracellular small ion permeability. They exhibit a relatively higher transepithelial resistance. This designation continues to be used to this day.

## 2. Strands, Pores, and Paracellular Permeability

A molecular basis for the paracellular barrier function was suggested by the finding that freeze fracture electron microscopy revealed arrays of anastomosing strands composed of small particles running through the tight junction region at the immediate subapical region of the plasma membrane of epithelial cells [8,9]. Multiple groups provided evidence that the leakiness of the paracellular permeability barrier was likely related to the structure/organization of these strands (see, e.g., [10,11]). Subsequent studies supported a correlation between tight junction strand number and epithelial tissue or monolayer transepithelial electrical resistance (TER), a measure of small ion permeability (see, e.g., [12,13]). This correlation was, however, not supported by further studies (see, e.g., [14]). These findings led to the proposal that tight junction permeability was mediated by a continuous barrier, comprised of these branching strands, punctuated by some number of openings of a defined size which allow for small ion permeability. It was predicted that the number of these openings (“pores”) varied between “tight” and “leaky” epithelia. This designation related solely to their relative permeability to small ions, the Pore Pathway, and was unrelated to permeability to larger molecules, the Leak Pathway (see below).

## 3. Tight Junction Structure

The tight junction is the structure which forms the epithelial paracellular permeability barrier and regulates the paracellular movement of solutes, including the paracellular movement of macromolecules. As a first step to understanding this process, many groups have begun to compile a list of proteins associated with the tight junction. The first protein shown to be associated with the tight junction structure was Zonula Occludens-1, ZO-1 [15]. ZO-1 is a cytoplasmic protein that binds to the cytoplasmic surface of some tight junction transmembrane proteins and links them to the cytoskeletal F-actin filaments [16,17]. Further studies identified two related proteins, ZO-2 and ZO-3, which are also associated with the tight junction structure cytoplasmic surface [18,19,20]. The first transmembrane protein found to be associated with the tight junction structure was occludin [21]. Occludin is a member of the MARVEL (MAL and related proteins for vesicle trafficking and membrane link) protein family that also includes Marvel D3 and tricellulin. These proteins, which form the Tight junction-Associated Marvel Protein (TAMP) subfamily, are all localized preferentially to the tight junction structure [22]. Occludin and Marvel D3 are localized preferentially to bicellular tight junctions (sites where two cells meet), whereas, tricellulin is localized preferentially to tricellular tight junctions (sites where three or more cells meet) [23]. Angulin proteins (angulin 1, angulin 2, and angulin 3) are also localized preferentially to tricellular tight junctions [24,25]. Junctional Adhesion Molecules (JAMs), members of the immunoglobulin superfamily of proteins, are localized preferentially to bicellular tight junction sites [26]. The first two members of the claudin family of transmembrane proteins (claudin-1 and claudin-2), which have been shown to be a major component of the tight junction strands (see below), were cloned in 1998 [27]. The claudin protein family has grown to include 27 members [28] that exhibit cell type-specific and tissue-specific expression patterns. In addition to this array of core tight junction proteins, a large number of cytoplasmic proteins have been shown to associate with the tight junction. These include cingulin [29] and an ever-expanding array of scaffolding and adaptor proteins, signaling proteins, cytoskeletal proteins, and regulatory proteins (see, e.g., [30,31]). The roles of these transmembrane proteins and cytoplasmic proteins in the structure, function, and regulation of the tight junction is an ongoing focus of study.

Tight junction proteins exhibit a range of homophilic and heterophilic binding interactions. Claudins exhibit both *cis* and *trans* interactions (see, e.g., [28,32,33]). Occludin exhibits *trans*-homophilic binding interactions [34]. Occludin, tricellulin, and Marvel D3 exhibit homophilic binding interactions within the same plasma membrane (*cis*-interactions). Marvel D3 also exhibits *cis*-heterophilic binding interactions with occludin and tricellulin. Some claudins exhibit *cis*-oligomerization with TAMPs. JAM-A exhibits homophilic *cis*-binding interactions [35] and homophilic *trans* binding interactions [36], as well as heterophilic binding interactions [37]. ZO proteins heterodimerize forming ZO-1/ZO-2 or ZO-1/ZO-3 dimers via their PDZ2 domains [16,19,38]. ZO proteins function as scaffolding proteins connecting the tight junction membrane proteins to the cytoskeleton and to various signaling proteins. The ZO proteins possess binding sites for many of the primary tight junction membrane proteins, including claudins (via the PDZ1 domain [38]), occludin (via the U5/GUK domain [16,39]), tricellulin [23], and JAM-A (via the PDZ3 domain and SH3 domain [40,41]). ZO proteins also possess binding sites for actin (via the Actin Binding Region), actin organizing proteins such as TOCA-1 (via the PDZ1 domain [42]), actin-binding proteins such as α-actinin-4 (via the PDZ-1 domain [43]) and cortactin (via a C-terminal domain [44]), and other cytoskeletal elements (see, e.g., [45,46]). These many-fold *cis* and *trans* binding events, both homophilic and heterophilic, support the hypothesis that the tight junction structure is a highly organized supramolecular complex of transmembrane and cytoplasmic proteins that is crosslinked to the actin cytoskeleton and, likely, to the microtubule network (see, e.g., [31,47,48]). Many studies have demonstrated that post-translational modifications of tight junction proteins, e.g., phosphorylation, can alter tight junction protein binding events and barrier function (see, e.g., [49,50,51,52,53,54]), suggesting a complex regulation of tight junction structure and function.

In addition to proteins, a role for lipids in forming the paracellular permeability barrier has been suggested (see, e.g., [55,56,57]). Of relevance to this review, cholesterol depletion in Caco-2 intestinal epithelial cells produced a dramatic increase in the paracellular permeability to large solutes, as well as to small ions [57]. A similar increase in paracellular permeability was observed under multiple conditions that disrupted the cholesterol-enriched lipid rafts both in cultured Caco-2 cells in culture and intestinal epithelia in vivo [58]. The ability of lipid composition manipulations to dramatically affect paracellular permeability both to small ions and solutes and to macromolecules suggests that the lipids play a central role for the overall maintenance of the paracellular permeability barrier.

These multiple binding interactions, along with much other data, have led to the proposal of a model of the tight junction structure. In this model, the tight junction permeability barrier is a highly crosslinked three-dimensional macromolecular lattice that forms a redundant steric barrier to paracellular solute movement (Figure 1). The claudins, via *cis* interactions, form antiparallel double row strands within the plasma membrane (see, e.g., [59,60]). The claudins within these strands interact vis *trans* interactions with complementary claudins in the claudin strands of apposing cell membranes [32,61]. These claudin strands form the primary barrier to paracellular movement of both small and large solutes, as well as, for pore-forming claudins, the pathway for small ion and small solute permeation through the tight junction, the Pore Pathway (see below). Once polymerized, these claudin structures are stabilized via binding to ZO protein dimers [62], as evidenced by low levels of claudin lateral mobility [63]. In addition to the claudin strand interactions joining adjacent epithelial cells, the TAMP’s (occludin, Marvel D3, and tricellulin) and JAM-A form *trans* homophilic adhesions between the membranes of adjacent epithelial cells. Occludin, Marvel D3, and JAM-A form these interactions in bicellular junctions. Tricellulin forms these interactions in tricellular junctions. Occludin exhibits a preferential, but not exclusive, localization to claudin strand branch points, which appear to be preferred sites for strand breaks [62]. All of these binding events serve to hold apposing epithelial cell membranes in close contact, thereby, limiting paracellular solute movement. Structural stability and flexibility of this tight junction macromolecular lattice is conferred via crosslinking of the tight junction membrane proteins to each other and to cytoskeletal elements, i.e., microfilaments and microtubules, mediated by ZO protein dimers and cingulin (see, e.g., [16,38,62,64,65,66]).

## 4. What Constitutes the Paracellular Permeability Barrier?

As described above, evidence supports the hypothesis that the tight junction strands are a central structure for the paracellular permeability barrier. There are at least three competing theories about what comprises the observed strands (see, e.g., [67]). As stated above, the leading theory at this time proposes the strands are comprised of strings of tight junction membrane proteins, primarily claudins. This hypothesis is supported by studies demonstrating the presence of strand-like structures in fibroblasts upon expression of multiple claudins (see, e.g., [68,69,70]). Co-expressing occludin with claudin-1 or claudin-2 resulted in the colocalization of claudins and occludin to these strands, consistent with both of these proteins being part of the tight junction structure [68]. Expression of either claudin-1 [71] or claudin-3 [72] in MDCK Type II renal epithelial (MDCK II) cells decreased paracellular permeability of both small and large solutes. The claudin-1 peptidomimetic, C1C2, increased permeability to both small and large molecules [73,74]. This was accompanied by internalization of tight junction claudins and occludin. Clostridium perfringens enterotoxin fragment binds to and downregulates claudin-3 and claudin-4, but not claudin-1 or claudin-2, from tight junction structures [75]. In MDCK II cells, the downregulation of claudin-4 from the membrane following treatment with the enterotoxin fragment was associated with a decrease in the number and complexity of tight junction strands and a loss of the paracellular permeability barrier to both small and large solutes. Expression of wild type claudin-5 in MDCK II cells decreased paracellular permeability to both large and small solutes [76]. In contrast, expression of mutants of claudin-5 which interfere with *trans* claudin-claudin interactions increased paracellular permeability to all solutes. Expression of these mutant claudin-5 proteins did not alter strand morphology, indicating that barrier disruption can be achieved by more subtle alterations in claudin-claudin interactions. Overexpression of Protein Kinase D in airway epithelial cells decreased claudin-1 expression [77]. This was paralleled by an increase in paracellular permeability to small and large solutes. A seemingly anomalous finding may provide further support for a central role for claudins in maintaining the paracellular permeability barrier. Expression of a c-myc-tagged claudin-1 in MDCK II cells increased permeability to some larger solutes. Expression of this construct, however, decreased small ion permeability [78]. This was paralleled by the appearance of tight junction-like adhesions deficient in occludin and ZO-1 that extended down the lateral membrane and exhibited an abnormal appearance in freeze fracture. Since claudin-1 is a barrier-forming claudin, the presence of a more basal belt of c-myc-claudin-1 would inhibit small ion permeability. The absence of ZO-1 and occludin in these aberrant strands might, however, alter the stability of the strands, thereby allowing permeation of large solutes. Consistent with this interpretation, a recent study demonstrated directly that the presence of occludin and tricellulin proteins in claudin-2 knockdown MDCK II cells, which express several other claudins, increased the complexity of the tight junction strands without changing the claudin protein expression pattern [79]. The cells with the less complex tight junction strand configuration exhibited decreased TER (increased Pore Pathway permeability) and increased permeability to macromolecules (Leak Pathway permeability). The increased Leak Pathway permeability was inversely related to molecular size. These data strongly support the hypothesis that claudin strands are a major component of the paracellular permeability barrier. Increased strand complexity is associated with decreased barrier permeability, at least for the Leak Pathway.

Further support for a central role for claudins is provided by a recent study which demonstrated that genetic knockout of five claudins expressed in MDCK II cells resulted in loss of the tight junction strands [80]. The study reported that the five claudin knockout cells exhibited a marked increase in the paracellular permeability to many solutes. The permeability increase was inversely correlated with solute size, as would be expected of a diffusive process. The importance of other tight junction membrane proteins in maintaining some level of permeability barrier is supported by the finding that, in the absence of the five claudins, JAM-A was essential to limit the paracellular permeability of larger macromolecules, possibly in conjunction with TAMPs [80]. 

A second theory proposes that the strands are lipid structures organized in the cylindrical hexagonal H_II_ phase [55,81]. These lipid structures form from the outer membrane leaflets of adjacent epithelial cells. Pure lipids in solution form micelles that exhibit intramembranous particles [82]. These particles may be located at sites of membrane fusion [83]. This theory remains controversial as one group reported the absence of diffusion of outer leaflet lipids between adjacent epithelial cells [84], which would appear to refute this theory, while another group reported outer leaflet lipid diffusion between adjacent epithelial cells [85]. It is somewhat difficult to reconcile the membrane fusion produced by the lipid structures, which might be expected to produce a relatively impermeant and static barrier, with the variable and highly regulated permeability to small ions and macromolecules observed in various epithelial tissues. While the absence of strands in the tight junction of the five claudin knockout cells seems to argue against a pure lipid basis for the tight junction structure, to date, this theory has not been definitively ruled out.

A third theory proposes a hybrid of the above two theories [86]. This theory proposes that the tight junction strands are comprised of a combination of the lipid cylindrical hexagonal H_II_ phase structures and protein components that mediate paracellular permeation across the lipid barrier. In support of this hypothesis, both lipids and tight junction proteins are co-localized to the tight junction structure (see, e.g., [86]). As described above, manipulation of membrane lipid composition has dramatic effects on the tight junction barrier function (see, e.g., [55,56,57,58]). Membrane fusion events, another circumstance in which cell membranes are brought into close apposition through the actions of proteins leading to changes in lipid membrane structure (see, e.g., [87,88]), may provide insights relevant to this theory for tight junction structure. The role of lipids in the tight junction structure and paracellular permeability barrier, either alone or with proteins, has been receiving increasing attention in recent years (see, e.g., [89]).

## 5. Two Paracellular Permeability Pathways: The Pore Pathway and the Leak Pathway

The possibility of multiple pathways mediating solute movement across epithelia was first suggested by Durbin et al. [90]. They reported the presence of two permeability pathways across frog gastric mucosa with pore sizes of 2.5 Å and 60 Å. van Os et al. [91] reported that rabbit gallbladder exhibited both an aqueous pathway for small electrolytes with a pore radius of ~4 Å and a second “shunt” pathway for larger nonelectrolytes with a radius of ~40 Å. These early studies, however, did not discriminate between transcellular and paracellular permeability pathways, raising questions about interpretation of the data. In an elegant series of experiments, Watson et al. [92] examined the paracellular permeability of polyethylene glycols (PEGs) of a range of sizes (Stokes radii—3.47–7.39 Å) in two intestinal epithelial cell lines, Caco-2 and T84. Plotting their results for PEG flux rate as a function of PEG radius revealed a biphasic behavior with a rapid decline in flux rate as PEG radius increased up to ~4 Å followed by a slower rate of decrease as PEG radius increased further. This biphasic behavior indicated the presence of, at least, two permeability pathways. The first pathway exhibited a pore radius of ~4.5 Å and a high capacity. This component was proposed to represent the high capacity, size- and charge-selective pathway for the paracellular movement of small ions and solutes, the Pore Pathway. Measurement of TransEpithelial Resistance (TER) under appropriate conditions is often used as a surrogate marker for Pore Pathway permeability since this is the major pathway for the paracellular permeation of small ions. The second pathway exhibited a low capacity and did not exhibit any solute size discrimination within their experimental setup; this pathway has come to be known as the Leak Pathway. While the study of Watson et al. [92] suggested that the Leak Pathway did not exhibit a size limit, it is possible that the largest solute examined in this study, 7.39 Å, was too small to evaluate this question adequately.

A third pathway for particles to cross the epithelial barrier has been described, the Unrestricted Pathway (see, e.g., [93]). This pathway is not associated with passage of compounds through the tight junctions. Rather, it results from epithelial damage/death, e.g., in Graft-Versus-Host Disease [94,95], that leads to gaps in the epithelial cell layer, allowing the transepithelial passage of particles including very large proteins, viruses, and bacteria. The Unrestricted Pathway is primarily observed in pathological states. Thus, it is fundamentally different from the Pore Pathway and Leak Pathway that mediate movement of solutes through the tight junction and are operative under physiological, as well as pathological conditions.

## 6. Pore Pathway Versus Leak Pathway: The Same or Different?

Studies from many groups have contributed to the consensus that the Pore Pathway permeability is mediated by a subset of claudins which bind to complementary claudins on the apposing membrane of adjacent epithelial cells to form size- and charge-selective pores through the tight junction. For example, MDCK C7 (Type I) renal epithelial cells do not express claudin-2 and exhibit a high TER. Expression of claudin-2, a cation pore-forming claudin, in these cells decreased TER (increased Pore Pathway permeability) without altering the permeability to mannitol or lactulose (larger solutes) [96]. This decrease in TER was due to an increase in tight junction cation permeability. Knockout of claudin-2 in MDCK II cells produced an increase in TER (decreased Pore Pathway permeability) but did not alter the permeability of larger solutes such as fluorescein and 4 kDa fluorescein-dextran (Leak Pathway permeability) [97]. For recent reviews of claudin structure and function, the reader is referred to [28,32,33]. These and much other data strongly support the hypothesis that the Pore Pathway allows the passage of small ions and solutes but not of macromolecules.

Barring the action of substrate-selective transporters, any opening that allows the passage of macromolecules will also allow the passage of small ions and small solutes. From this, it must be concluded that the Leak Pathway should mediate the passage of both macromolecules and small ions and solutes. Consistent with this conclusion, many studies have reported parallel changes in Pore Pathway permeability and Leak Pathway permeability in response to different modulators in a variety of epithelia (see, e.g., [98,99,100,101,102]. Multiple studies, however, have demonstrated increased permeability to large solutes without a parallel increase in small ion or small solute permeability, measured typically by TER. Using the PEG size profiling technique described above [92], it was shown that treatment of T84 cells with interferon-γ increased Leak Pathway permeability while not affecting Pore Pathway permeability [103]. Treatment of MDCK II cells with interferon-γ and tumor necrosis factor-α both increased TER (decreased Pore Pathway permeability) and increased the paracellular flux of 3 kDa fluorescein-dextran (increased Leak Pathway permeability) [104]. Mercado et al. [105] examined the effects of five nutraceuticals (zinc, quercetin, indole, butyrate, and nicotine) on Pore Pathway and Leak Pathway permeability in the LLC-PK_1_ renal epithelial cell line. They found no consistent concordance in the effects of these compounds on the permeabilities of the Pore Pathway and the Leak Pathway. In one study, RhoA activation, a small G protein that regulates actin organization, both increased TER (decreased Pore Pathway permeability) and increased mannitol flux (increased Leak Pathway permeability) in MDCK II cells [106]. A second study, however, reported increases in both Pore Pathway and Leak Pathway permeability upon expression of a constitutively active RhoA mutant in MDCK II cells [107]. We have shown that treatment of both MDCK II cells and LLC-PK_1_ cells with low concentrations of hydrogen peroxide increased Leak Pathway permeability without altering TER [108]. The ability to manipulate Leak Pathway permeability without a parallel change or even with an opposite change in Pore Pathway permeability strongly argues that the Leak Pathway is a distinct paracellular permeability pathway.

## 7. What Is the Leak Pathway?

Despite increasing attention in recent years (see, e.g., [67]), the basis for permeation via the Leak Pathway is still unclear. Over the years, several theories have been proposed.

Cell Damage/Death Model: One hypothesis is that the Leak Pathway represents sites of epithelial cell damage/death where a cell detaches from the surrounding cells in the epithelial sheet. This would allow the permeation of both large and small solutes but, if the damage/death is rare within the cell layer, the permeation of small ions and solutes might be insufficient to alter the global flux rate for small ions and solutes while allowing some level of macromolecule permeation. The ability to regulate reversibly Leak Pathway permeability without evidence of cell damage/death, however, seems inconsistent with this hypothesis. For example, we were unable to demonstrate a change in cell damage/death via multiple pathways under conditions that produced an increase in Leak Pathway permeability, but not Pore Pathway permeability, of MDCK II cells [108]. Treatment of MDCK II cells with interferon-γ and Tumor Necrosis Factor-α increased Leak Pathway permeability while not affecting Pore Pathway permeability [104]. This treatment did not increase cell damage/death in these cells [109]. As described above, this model appears to represent the Unrestricted Pathway described by France and Turner [93] that is observed primarily in pathogenic conditions such as Graft-Versus-Host Disease [94,95].

Tricellular Junction Model: Another hypothesis is that the Leak Pathway is localized to the tricellular junction, composed of tricellulin and angulin family proteins [110]. Moderate overexpression of tricellulin in MDCK II cells, which express low levels of endogenous tricellulin, decreased the paracellular permeability of both 4 kDa fluorescein-dextran and 10 kDa fluorescein-dextran [110]. The authors reported minimal permeability of MDCK II cells to 20 kDa fluorescein-dextran and larger macromolecules. Knockdown of tricellulin protein content in HT-29/B6 intestinal epithelial cells, which express higher endogenous tricellulin levels, increased the paracellular permeability of both 4 kDa fluorescein-dextran and 10 kDa fluorescein-dextran [111]. The results suggest that the tricellular junction forms a paracellular channel/pore for molecules with Stokes radii up to between 23 Å (10 kDa fluorescein-dextran) and 33 Å (20 kDa fluorescein-dextran). Tricellulin limits the permeability via the tricellular junction pores. These results are consistent with previous ultrastructural analysis which indicated that the tricellular junction formed a pore of ~5 nm (~50 Å) radius and ~1 μm length [112].

Some results, however, are inconsistent with the hypothesis that the tricellular junction can completely account for Leak Pathway permeability. Multiple studies have reported that paracellular permeability does not drop off dramatically for solutes larger than 20–30 Å (see, e.g., [71,100,113]). The permeability of solutes with radii greater than this threshold would be greatly limited by a pore of the tricellular junction dimensions. Knowing the dimensions of the tricellular pore shown above, one can calculate the maximal permeability of the 4 kDa or 10 kDa fluorescein-dextrans through the tricellular pores using a modification of the Stokes-Einstein equation that accounts for the presence of a limited number of openings of fixed size.
P(r) = (ε/δ)(k_B_T/6πηr)
where P(r) is the apparent permeability, ε is the fractional pore area, δ is the pore length, r is the solute Stokes radius, k_B_ is the Boltzmann’s constant, T is the temperature, and η is the viscosity of the solution.

From direct observation, the number of tricellular junctions per microscopic field of post-confluent MDCK II cells averaged 35 tricellular junctions/3600 μm^2^ cell surface area.
Area of tricellular openings = (tricellular opening area) × (number of tricellular openings)= π(50 Å)^2^ × 35= 274,889 Å^2^
ε = (tricellular opening area)/(total surface area)= (27.4889 × 10^4^ Å^2)^/(36 × 10^10^ Å^2^)= 7.636 × 10^−7^
δ = 10^4^ Å
η = νρ
where ν is the measured kinematic viscosity and ρ is the density (assumed to be 1 g/cm^3^).
= 0.9644 × 10^−10^ g/Ås
k_B_ = 1.38 Å^2^g/s^2^K
T = 310° K
r = 14 Å (4 kDa fluorescein-dextran Stokes radius)
P(r)_calculated_ = 1.283 Å/s

P(r)_measured_ = 3.7125 Å/s for 4 kDa fluorescein-dextran across MDCK II cell monolayers.

This calculation does not assume any hindrance of solute passage, some level of which would be expected for these solutes passing through pores with a radius of 50 Å, which would decrease the permeability from this maximal rate. This theoretical upper limit for the solute permeability yielded a P_app_ that was approximately 30% of the measured P_app_. Thus, the tricellular pores cannot account for even a majority of the permeation of these solutes across MDCK II cell monolayers. They cannot constitute the entirety of the Leak Pathway, although they may contribute significantly to the permeability of some solutes under certain conditions.

Strand Break Model: A third hypothesis is that Leak Pathway permeability is mediated by transient localized breaks in the tight junction barrier. Since this barrier appears to be formed by the anastomosing strand network, the macromolecules would traverse the tight junction by passing through sequential breaks in individual strands until the macromolecule emerges on the other side of the tight junction (see, e.g., [31,114]). As indicated above, fibroblasts transfected with claudins exhibit strand-like structures [68,69,70]. Transient breakage and resealing of these strands was observed [62,115]. Discontinuities in the strand network have been observed in cultured MDCK II cells [116] and in intestinal epithelia in vivo [117,118], supporting their existence in intact epithelia. If the spaces between successive strands in freeze fracture images represent tiny pockets of aqueous solution, which has not been shown to date, a macromolecule could pass from one aqueous compartment to the next through a transient strand break much as a boat passes through a series of locks [31,92,119]. As described by Tervonen et al. [114], this model makes multiple testable predictions about parameters, such as opening dynamics, that would affect Leak Pathway permeability.

A recent study, however, suggests an added complexity to the paracellular permeability to macromolecules, i.e., Leak Pathway permeability. As described above, Otani et al. [80] knocked out expression of five claudins in MDCK II cells. This resulted in the loss of the tight junction strands with a parallel increase in small ion permeability. Despite the loss of tight junction strands, the subapical membrane regions of adjacent cells remained closely apposed. Permeability to macromolecules was increased, but the extent of increase decreased progressively with increasing macromolecule size. This suggests that another component(s) of the tight junction contributes to maintaining close cell–cell appositions that form a partial barrier to the paracellular passage of larger macromolecules. In this study [80], knockout of JAM-A alone did not substantially alter paracellular permeability to either small ions or macromolecules, but JAM-A knockout in the five claudin knockout cells resulted in a dramatic increase in paracellular permeability to larger macromolecules. The authors concluded that JAM-A, either alone or in combination with TAMP proteins, can form a crude barrier to larger macromolecule permeability in the absence of the tight junction strands. While these findings do not refute the strand break hypothesis, they suggest a further complexity to what forms the paracellular permeability barrier. The “strand break” model also does not directly address the question of what causes these strand breaks and how they are regulated.

Strand Break-Tension Model: As discussed below, multiple studies have indicated that altered tension on the tight junction can increase Leak Pathway permeability. This suggests an enhancement of the “strand break” model to incorporate altered tension on the tight junction as a mechanism through which strand breaks are induced. Altered tension on the claudin strands, either due to altered cytoskeletal force or to decreased structural stability, would lead to localized breaks in the strands with a resultant increase in Leak Pathway permeability. If the alterations are global in nature, this would result in detectable increases in the permeability to both small and large molecules. If the alterations are more localized, this could result in increases in the permeability to macromolecules but, possibly, without a measurable increase in small ion and solute permeability. The report that the ZO protein-actin binding is a relatively weak interaction [120] could provide flexibility in the regulation of cytoskeleton-induced imposition of tension on the tight junction structure.

Within the “strand break” model, one could imagine two extremes to how this might function. In one extreme, the paracellular passage of a macromolecule across the tight junction structure would be through its progressive movement through intra-tight junction aqueous compartments via successive strand breaks, as described above (Figure 2a). Each of these strand breaks could be the result of highly localized changes in tension on the tight junction structure. The alternative possibility is that tension produces a localized disruption of the entire tight junction structure such that an opening is created connecting to the two bulk fluid compartments (Figure 2b). These two possible scenarios would produce different kinetic behaviors for the appearance of the macromolecules in the receiver fluid compartment. In the first scenario, the macromolecules would appear in a burst-like pattern. The bursts might appear repeatedly at the same sites or might appear at multiple sites within the cell monolayer. In the second scenario, the macromolecules would emerge continuously for an extended period of time at the same site until the opening was closed.

## 8. What Cell Components Are Part of the Leak Pathway?

While an understanding of exactly what is the Leak Pathway remains unclear, many studies have identified components that, when manipulated in some fashion, can modulate Leak Pathway permeability. An exhaustive discussion of this topic is not possible due to the substantial literature related to this topic. We will, instead, attempt to provide examples of the different implicated components and the various manipulations of them that modulate Leak Pathway permeability.

Claudins: Claudins appear to be central to the formation of a paracellular permeability barrier and the small ion- and small solute-selective pathways across this barrier (for reviews, see, [28,93,121,122,123]). As described above, manipulation of claudin content alters paracellular permeability to both small ions and macromolecules in predictable manners. In addition, Inai et al. [71] reported that constitutive expression of c-myc-tagged wild type claudin-1 in MDCK II cells increased TER and decreased Leak Pathway permeability, consistent with the barrier-forming function of claudin-1 (see, e.g., [124]). As mentioned above, however, McCarthy et al. [78] reported that inducible expression of a c-myc-tagged claudin-1 in MDCK II cells increased TER but also increased Leak Pathway permeability. This was accompanied by the appearance of abnormal tight junction strands. McCarthy et al. [78] suggested that the difference in results between the two studies may be due to differences in the expression levels of ZO-1 observed upon constitutive versus inducible expression of c-myc-tagged claudin-1. Mutations in a conserved claudin protein domain in two different claudins (mouse claudin-15 and mouse claudin-14), which are predicted to disrupt claudin-claudin interactions, disrupted strand organization and increased the number of strand discontinuities [69]. The effect of expressing the mutated claudins on paracellular permeability to either small ions or macromolecules was not reported.

Post-translational modification of claudins that alter claudin content or claudin-protein interactions have also be reported to regulate paracellular permeability. Claudin phosphorylation has been demonstrated in multiple studies (see, e.g., [54]). In OVCA433 ovarian cancer cells and in MDCK I cells, expression of a mutant claudin-3 (barrier-forming claudin) that mimics cAMP-mediated phosphorylation of claudin-3 on threonine 192 (T192D) decreased TER (increased Pore Pathway permeability) and increased the permeability to 4 kDa fluorescein-dextran [125]. These effects may be explained by the observation that the mutant claudin-3 appeared to be more diffusely organized at the tight junction. Phosphorylation of claudin-2 (cation pore-forming claudin) at tyrosine 6 in the PDZ1 domain disrupted its interaction with ZO-1 and decreased its localization to the tight junction [126]. The TER was increased in cells expressing the phosphorylated claudin-2 but its effect on Leak Pathway permeability was not examined. Treatment of MDCK II cells with Hepatocyte Growth Factor increased the dynamic mobility of claudin-3, measured using Fluorescence Recovery After Photobleaching (FRAP), and decreased tight junction integrity [127]. It was suggested that the increased dynamic mobility represented decreased incorporation of claudin-3 into the tight junction macromolecular complex. These results suggest that post-translational modifications of claudins may be capable of regulating the paracellular permeability barrier to macromolecules.

Tricellulin: As described above, studies have indicated that manipulation of tricellulin content can alter permeability to, at least some, large solutes in multiple epithelial cell types (see, e.g., [110,111]). The extent of the tricellular pore contribution to overall Leak Pathway permeability may depend on the relative macromolecule permeability of the epithelial cell type. It may contribute a greater proportion of the total permeability in epithelial cell types exhibiting a lower total Leak Pathway permeability.

Occludin: Occludin preferentially, but not exclusively, localizes to strand branch points which appear to be a major locus for strand breaks [62]. Occludin, along with tricellulin, increase strand complexity in MDCK II cells [79]. Thus, manipulation of occludin content or organization could modulate Leak Pathway permeability. Many studies, however, reported that manipulation of occludin protein content did not alter Leak Pathway permeability. Schulzke et al. [128] reported no effect of occludin knockout on intestinal mannitol flux. Epithelial cells derived from embryonic stem cells in which occludin expression was knocked down did not exhibit a major change in permeability to NHS-LC-biotin [129]. Since NHS-LC-biotin is a linear molecule, it is unclear which permeability pathway this solute would measure. Several groups have reported no effect of occludin knockdown on Leak Pathway permeability in MDCK II cells [79,130,131,132]. Occludin knockdown in Caco-2 cells was reported to increase Leak Pathway permeability [100,133], although this was later disputed [134]. Occludin overexpression did not alter Leak Pathway permeability in MDCK II cells [104,130]. Two studies [135,136] reported that expression of chicken occludin in MDCK II cells increased Leak Pathway permeability. It is possible that the chicken and endogenous dog occludin proteins were not entirely compatible raising questions about the interpretation of this result. Overall, these studies are consistent with the hypothesis that simply manipulating occludin protein content is not sufficient to alter Leak Pathway permeability.

The presence or absence of occludin may not directly alter Leak Pathway permeability. It may, however, alter the stability of the claudin strands, making them more or less susceptible to disruption upon imposition of tension/stress on the macromolecular tight junction structure. Consistent with this proposal, the ability of hydrogen peroxide to increase Leak Pathway permeability in MDCK II cells was enhanced by occludin knockdown and diminished by occludin overexpression [108]. Hydrogen peroxide treatment slowed the dynamic mobility of occludin protein [108], suggesting an alteration in the tight junction structure. Several studies suggest that treatment with hydrogen peroxide will increase stress/tension on the tight junction structure. Hydrogen peroxide treatment increased motility of multiple epithelial cell types (see, e.g., [137,138,139]), including Caco-2 cells [140]. These results suggest that hydrogen peroxide may increase tension at the cell–cell junctions that, in occludin knockdown cells, may be more susceptible to localized disruptions and increased Leak Pathway permeability. Consistent with this suggestion, overexpression of occludin protein in mouse intestine diminished the ability of Tumor Necrosis Factor (TNF) to induce leak of bovine serum albumin across the intestinal epithelium [141]. Knockout of occludin in mouse intestine, however, diminished the ability of EGTA (calcium chelator) to increase Leak Pathway permeability [142]. This was associated with a diminished ability of EGTA to induce internalization of tight junction and adherens junction proteins. Van Itallie et al. [104] reported that overexpression of occludin enhanced and occludin knockdown diminished the ability of cytokines, TNF-α and interferon-γ, to alter TER and the permeability of large solutes in MDCK II cells. Using cell motility as a surrogate for cell–cell junction stress, interferon-γ has been reported to increase the motility of airway epithelial cells [143] but to decrease the motility of T84 cells [144]. This suggests the effect of a specific stimulus on tight junction tension/stress may be cell type specific.

Studies examining the effect of expression of occludin mutants or protein fragments on paracellular permeability also provide support for a role in regulating Leak Pathway permeability. The occludin COOH-terminal cytoplasmic tail has been shown to mediate inter-protein interactions as well as being involved in targeting occludin to the tight junction and development and maintenance of barrier function (see, e.g., [39,135,145,146]). A COOH-terminal domain deletion mutant of occludin protein expressed in MDCK II cells exhibited a discontinuous distribution within the tight junction, in contrast to the typical continuous distribution [135], suggesting a disruption of the normal tight junction structure. Cells expressing this mutant occludin protein exhibited a dramatically increased paracellular flux of 4 kDa fluorescein-dextran. The Leak Pathway permeability increase was proportional to the level of expression of this mutant occludin protein. A mutant occludin protein, in which portions of the first or second extracellular loop were deleted, expressed in MDCK II cells exhibited a continuous distribution that extended down the lateral membrane, suggesting an enhanced tight junction structure [145]. Cells expressing this mutant occludin protein exhibited decreased paracellular permeability to mannitol and horseradish peroxidase. Addition of a synthetic peptide corresponding to the second extracellular loop of occludin protein to *Xenopus* renal epithelial cells (A6) increased occludin turnover and, thereby, decreased occludin content in the tight junction [146], suggesting disruption of the tight junction structure. Cells treated with this peptide exhibited dramatically increased paracellular permeability to multiple solutes, i.e., mannitol, inulin, 3 kDa fluorescein-dextran, and 40 kDa fluorescein-dextran [146]. This effect was not observed with synthetic peptide corresponding to the first extracellular loop.

Many studies have demonstrated the post-translational modification of occludin and its association with modulation of paracellular permeability, including Leak Pathway permeability (for reviews, see, [49,147,148,149]). In general, tyrosine phosphorylation of occludin is associated with increased paracellular permeability (see, e.g., [150,151,152,153,154]) and serine/threonine phosphorylation of occludin is associated with decreased permeability (see, e.g., [53,155,156]). Many, but not all, of these studies examined both Pore Pathway and Leak Pathway permeability. For example, Manda et al. [142] reported that expression of a deletion mutant occludin protein lacking a C-terminal sequence containing multiple serine, threonine, and tyrosine phosphorylation sites, called the Occludin Regulatory Motif (ORM), decreased occludin protein dynamic mobility in MDCK II cells and in IEC6 intestinal epithelial cells. Deletion of this sequence attenuated the abilities of multiple stressors, including calcium depletion, osmotic stress, and hydrogen peroxide, to disrupt the tight junction structure and increase paracellular permeability via both the Pore Pathway and the Leak Pathway. Protein Kinase C-ζ-mediated phosphorylation of occludin on serine/threonine residues inhibited the increase in Pore Pathway permeability caused by *Staphylococcus aureus* infection of airway epithelial cells [157]. Leak Pathway permeability was not examined in this study.

In aggregate, these results support a role for occludin in regulating Leak Pathway permeability. Further investigation of these observations is required to clarify the basis for the apparent regulatory action. The preferential localization of occludin to strand branch points, which are also preferential sites for strand breaks, suggests one possible mechanism through which occludin could regulate Leak Pathway permeability.

Junctional Adhesion Molecules: Knockdown of JAM-1 (JAM-A) in SK-CO15 colonic epithelial cells increased permeability via both the Pore Pathway (decreased TER) and the Leak Pathway (increased permeability to 4 kDa fluorescein-dextran) [158]. Consistent with this result, the intestinal permeability to both small ions and 4 kDa fluorescein-dextran was increased in JAM-A knockout mice [159]. Both systems also showed increased permeability to larger solutes up to 40 kDa fluorescein-dextran [160]. In contrast, knockout of JAM-A in MDCK II cells did not affect either Pore Pathway or Leak Pathway permeability [80]. As described above, the five claudin knockout MDCK II cells exhibited an increase in small ion permeability and an increase in Leak Pathway permeability that diminished as molecular size increased. Further knockout of JAM-A in these cells resulted in a loss of the size selectivity to macromolecule permeability. These results suggest a complexity to the role of JAM-A in the regulation of Leak Pathway permeability that may reflect tissue-specific differences.

ZO Proteins: The ZO proteins crosslink the tight junction membrane proteins to create a macromolecular lattice and connect them to the underlying actin cytoskeleton [16,38,39,40,41,45,62,161,162]. Knockdown of ZO-1 protein expression in MDCK II cells increased selectively the permeability to large solutes (Leak Pathway), including polyethylene glycol, mannitol, and 3 kDa fluorescein-dextran [45]. Consistent with these findings, Tokuda et al. [163] reported that complete knockout of ZO-1 protein expression in MDCK II cells had no consistent effect on TER but produced a variable increase in permeability to 4 kDa fluorescein-dextran. Van Itallie et al. [45] reported that knockdown of ZO-2 protein expression in MDCK II cells, in contrast, did not affect either TER or permeability to large solutes. We have confirmed an effect of ZO-1 knockdown but not of ZO-2 knockdown on Leak Pathway permeability in MDCK II cells [132]. Hernandez et al. [164], however, reported that knockdown of ZO-2 protein expression in MDCK II cells increased paracellular permeability to 70 kDa fluorescein-dextran but did not affect TER. Raya-Sandino et al. [113] reported that knockdown of ZO-2 protein in MDCK II cells decreased paracellular permeability to 10 kDa fluorescein-dextran and 70 kDa fluorescein-dextran and increased TER. This was accompanied by upregulation of ZO-1 and claudin-4 expression and down-regulation of claudin-2 and paracingulin expression, possibly providing an explanation for this apparently anomalous result. Double knockdown of both ZO-1 and ZO-2 protein expression in MDCK II cells did not alter TER but dramatically increased the paracellular flux of 3 kDa fluorescein-dextran [161]. Re-expression of full-length ZO-1 protein in these double knockdown MDCK cells restored the flux of 3 kDa fluorescein-dextran to near wild type levels. In the ZO-1/ZO-2 double knockdown MDCK cells, expression of ZO-1 mutant proteins containing targeted deletions of specific protein domains revealed that the PDZ1, PDZ2, SH3, and U5 domains were most critical for ZO-1 to limit Leak Pathway permeability [165]. Spadaro et al. [166] reported that ZO-1 protein displays tension-dependent transitions between a stretched and a folded conformation that modulates exposure of protein binding sites. It was suggested that ZO-1, possibly through this folded/stretched transition, could regulate cytoskeletal tension on the tight junction structure [167]. Thus, ZO-1 depletion could lead to a less stable tight junction structure. These results suggest that ZO proteins have a major role in control of Leak Pathway permeability. The results are consistent with the hypothesis that diminishing or eliminating the link between the actin cytoskeleton and the tight junction membrane proteins would destabilize the paracellular permeability barrier to macromolecules but not necessarily to small ions and solutes.

Actin Cytoskeleton: Many studies support a role for the actin cytoskeleton in regulation of epithelial barrier function (see, e.g., [168,169,170]). One of the most well-studied examples is the role of myosin light chain kinase (MLCK) in regulating intestinal epithelial cell permeability (for reviews, see [102,171]). MLCK-mediated contraction of the perijunctional actomyosin ring regulated Caco-2 cell and intestinal tissue paracellular permeability under physiological conditions (see, e.g., [172]) and pathophysiological conditions (see, e.g., [157,173]). Under physiological conditions, this leads to an increase in the paracellular permeability to small ions and solutes but not to larger molecules, whereas, in the pathophysiological states, this leads to increased permeability to both small solutes and macromolecules. The perijunctional actomyosin ring contraction induces changes in tight junction organization and protein content. Studies in other systems also suggest a role for the actin cytoskeleton in the regulation of Leak Pathway permeability. Van Itallie et al. [42] reported that ZO-1 binds to TOCA-1, a BAR-domain containing protein involved in formation of branching F-actin networks [174], and targets TOCA-1 protein to the tight junction. Knockout of TOCA-1 protein expression in MDCK II cells produced an increase in the permeability of 3 kDa fluorescein-dextran while not affecting TER [42], similar to the effect of ZO-1 knockdown or knockout in these cells. Further supporting a role for F-actin branching networks in the regulation of Leak Pathway permeability, treatment of MDCK II cells with the Arp2/3 inhibitor, CK666, increased the permeability to 3 kDa fluorescein-dextran and this effect was synergistic with TOCA-1 knockout [42]. In addition, treatment of either ZO-1 knockdown [45,132] or TOCA-1 knockout [118] MDCK II cells with blebbistatin, which inhibits myosin ATPase activity, increased paracellular permeability to large solutes. In contrast, treatment of wild type MDCK II cells with blebbistatin had no effect on permeability of either calcein or 3 kDa fluorescein-dextran [45,132]. These results support a central role for the actin cytoskeleton in regulating tight junction integrity and Leak Pathway permeability. They suggest this regulation is mediated, at least in part, through F-actin association with the tight junction via ZO-1.

Small G Proteins: Small GTPases, Rho, cdc42, and rac, regulate actin cytoskeletal dynamics to modulate multiple cellular processes (for reviews, see, e.g., [175,176,177]), including regulation of the tight junction structure and function (for reviews, see, e.g., [175,178,179]). RhoA activation [106] and expression of constitutively active RhoA [107,180], rac1 [180], or cdc42 [180] in MDCK II cells increased Leak Pathway permeability. The effects on TER were variable. Interestingly, expression of dominant negative mutants of RhoA, rac1, and cdc42 in MDCK II cells produced similar increases in paracellular permeability [180], suggesting that maintaining the correct levels of RhoA, rac1, and cdc42 protein activation are critical for maintaining barrier function. Treatment of T84 cells with *Escherichia coli* cytotoxic necrotizing factor-1 (CNF-1) activated RhoA, rac1, and cdc42 and increased paracellular permeability to both small ions and macromolecules [181]. Changes in paracellular permeability were associated with internalization of tight junction proteins and/or alterations in the actin cytoskeleton. In Caco-2 cells, knockdown of the GTPase exchange factor (p114RhoGEF), which exchanges GDP for GTP on RhoA leading to RhoA activation, inhibits recovery of paracellular barrier function, both Pore Pathway and Leak Pathway, during a calcium depletion/repletion maneuver [182]. Stephenson et al. [183] reported a role for RhoA activation in repairing localized sites of tight junction disruption. Sites of paracellular leak were produced by junction elongation as cells within the *Xenopus laevis* gastrula divided. This likely increased tension on the tight junction structure at these sites, which exhibited dilution of tight junction proteins, ZO-1 and occludin, at the site of membrane elongation. Repair was initiated by localized RhoA activation, which was followed by localized actin polymerization, membrane contraction, and recruitment of ZO-1, occludin, and claudin-6 proteins to the breach site. Recruitment of these proteins led to reinforcement of the tight junction structure and repair of the paracellular permeability barrier breach.

Reflecting on apical junctional integrity, it is interesting to note the marked parallels between regulation of tight junction and adherens junction integrity. These include junctional tensile stress (tight junction—[183]; adherens junction—[184]), localized RhoA activation (tight junction—[170,183]; adherens junction—[18]), p114RhoGEF activation (tight junction—[182]; adherens junction—[184]), Gα12 activation (albeit in opposite directions; tight junction—[185]; adherens junction—[184]), mDia1 organization of actin structures (tight junction and adherens junction—[186]), and a tension sensor which transitions between a closed and an open conformation that exposes protein binding sites (tight junction (ZO-1)—[166]; adherens junction (α-catenin)—[187]). These similarities could represent a convergent evolution of regulatory mechanisms for the two junctional structures or, alternatively, an overlap in the mechanisms regulating the two junctional structures.

## 9. Does Tight Junction Stress/Tension Affect Leak Pathway Permeability?

Multiple studies indicate that imposition of external stress that would be predicted to increase tension on the tight junction structure increases paracellular permeability. Axial stretching of alveolar epithelial cells in culture increased paracellular permeability to both small and large molecules [188,189,190,191]. The change in permeability was only observed at the highest level of stretch, suggesting a threshold behavior. The increased alveolar epithelial cell paracellular permeability was paralleled by loss of occludin protein [189,190]. It was concluded that stretch increased the number of large openings without affecting opening size in these cells [188,189]. Cyclic stretch of Caco-2 cells increased the paracellular permeability to FITC-inulin (Leak Pathway) [192]. This was paralleled by a change in the tight junctions from a more linear to a wavy configuration and subsequent internalization of tight junction proteins. Imposition of an osmotic gradient across a Caco-2 cell monolayer produced an increase in both Pore Pathway and Leak Pathway permeability [193]. This was paralleled by a redistribution of occludin and ZO-1 proteins from the tight junction region and disruption of the actin cytoskeletal organization. Cattaneo et al. [194] attributed collapse of domes in MDCK II cell cultures exposed to shear stress to increased paracellular permeability although they did not measure paracellular permeability directly.

Studies have demonstrated a role for internal tension in the regulation of the adherens junction (see, e.g., [184,195,196]). Based on the similarities in the two junctions described above, it is logical to hypothesize that internal tension will also regulate tight junction integrity and paracellular permeability, including Leak Pathway permeability. Multiple studies have demonstrated that under baseline conditions, the tight junction structure is under tension. Haas et al. [197] demonstrated directly, using a ZO-1 tension sensor, that the ZO-1 protein located at the tight junction is under tensile stress. The level of tensile stress was modulated by crosstalk between the tight junction and the cell-substratum interactions. This was mediated by JAM-A, p114RhoGEF, and RhoA. Knockout of TOCA-1 in MDCK II cells both decreased tight junction tension and increased Leak Pathway permeability [42]. Leak Pathway permeability in these cells was increased further by inhibiting myosin ATPase activity [42,132], which may further relax tension on the tight junction although this was not examined directly. Double knockdown of ZO-1 and ZO-2 in MDCK II cells produced an expansion of the cortical actin ring and decreased the apical cross-sectional area [161]. This was associated with an increased Leak Pathway permeability without a change in Pore Pathway permeability. This has similarities to, but is not identical to, the myosin light chain kinase-mediated contraction of the cortical actin ring observed in intestinal epithelial cells [157,168,169,170,171,172,173]. In Caco-2 cells, knockout of the formin, mDia1, an actin-binding protein that promotes the formation of parallel F-actin networks, increased the paracellular permeability to both small ions and 4 kDa fluorescein-dextran [187]. This was associated with decreased total content and increased mobile fraction of occludin and ZO-1. Similar effects were observed when wild type Caco-2 cells were treated with the Rho-associated protein kinase (ROCK) inhibitor, Y-27632 [186]. In the *Xenopus laevis* gastrula, tricellular junctions, sites of enhanced junctional tension [198], exhibited a somewhat higher incidence of tight junction breaches compared to bicellular junctions [86]. As described above, junctional elongation occurring during cytokinesis in these gastrulas caused dilution of specific tight junction proteins, occludin and ZO-1, but not of claudin-6. These localized sites of occludin and ZO-1 depletion exhibited transient increases in Leak Pathway permeability. Junctional contraction, mediated by RhoA-initiated actomyosin recruitment and increased contractility, “repair” these transient breaches in the paracellular permeability barrier.

It is important to note that changes in Leak Pathway permeability have been noted under conditions that both increase and decrease tight junction tensile stress. One possible and, at present, highly speculative explanation relates to the direction of the tensile stress being modulated. The cortical actin ring might tug on tight junction proteins laterally along the membrane, i.e., parallel to the plane of the cell membrane. In contrast, the branching actin network and actin stress fibers might tug on tight junction membrane proteins in the direction of the cell body, i.e., perpendicular to the plane of the cell membrane. It is currently unclear how changes in tension in these different directions might affect tight junction integrity and paracellular permeability. Differences may also relate to the level of imposed stress. Lower levels of tension might induce minor damage to the tight junction structure that triggers repair mechanisms that actually enhance the tight junction structure and, thereby, decrease paracellular permeability. Higher levels of tension might induce more substantial damage that cannot be repaired sufficiently leading to increased paracellular permeability. Alternatively, the tight junction might be under optimal tension under normal conditions. Either increasing or decreasing the level of tension might lead to disruption of tight junction barrier integrity. Clearly, much more work needs to be done to investigate how stress on the tight junction structure affects paracellular permeability, in particular the Leak Pathway.

## 10. How Can Leak Pathway Permeability Be Regulated Independently from Pore Pathway Permeability?

Since openings that permit the passage of large solutes must also allow the passage of small solutes, the results presented above raise the question of why increased Leak Pathway permeability is not always paralleled by increased Pore Pathway permeability. This apparent inconsistency can be reconciled by considering two different scenarios. In the first scenario, there is global disruption in the paracellular permeability barrier of most, if not all, cells within the cell population. This would result in a dramatic increase in the permeability of the epithelial barrier to both small ions and macromolecules. This scenario would be consistent with the effects of experimental conditions that are reported to increase permeability via both the Pore Pathway and the Leak Pathway. In this case, however, the increased permeability to small ions would not be primarily via the claudin pores (classic Pore Pathway) but, rather, would reflect the greater movement of small ions through the extensive and larger openings of the Leak Pathway. Consistent with this hypothesis, it was reported that treatment of HT-29/B6 cells with Tumor Necrosis Factor-α, which increases permeability to both small ions (Pore Pathway) and macromolecules (Leak Pathway), markedly increased the number of sites of macromolecule passage within the cell monolayer [199]. 

In the second scenario, there are relatively rare, highly localized barrier disruptions between some cells within a cell population. This would lead to a detectable increase in the permeability to macromolecules (Leak Pathway) since permeability to macromolecules is very low under normal conditions. For epithelial cell populations exhibiting a relatively high level of Pore Pathway permeability under normal conditions, however, the relatively minor contribution of small ions and molecules moving through a limited number of highly localized sites of barrier disruption might not be evident on top of this high “background” Pore Pathway permeability. Epithelial cells exhibiting a low Pore Pathway permeability may be more likely to exhibit an increase in Pore Pathway permeability from these rare barrier disruptions, but this remains to be determined.

Two recent studies demonstrating a small number of discrete sites of macromolecule passage within an epithelial cell population support the hypothesis that these can be rare events [183,199]. Stephenson et al. [183] reported that sites of large solute permeation across the epithelial cell layer in the *Xenopus laevis* gastrula were discrete, limited in number, and transient, remaining open for no more than 5 min. Richter et al. [199] reported that macromolecule permeability across monolayers of MDCK II cells, Caco-2_BBE_ cells, and HT-29/B6 cells was localized to specific transient sites, being “open” for 30 min or less. At any time, small regions around only some cells within an epithelial cell population exhibited macromolecule permeation. The locations of these permeation sites changed over time.

## 11. TAMPs, Tight Junction Structure, and Leak Pathway Permeability

It is worth returning briefly to the TAMPs and their potential involvement in maintaining the tight junction structure and regulating Leak Pathway permeability. As described above, there are currently three TAMPs, occludin, tricellulin, and Marvel D3. Under normal conditions, both occludin and Marvel D3 are localized primarily to bicellular junctions, whereas, tricellulin is localized primarily to tricellular junctions. At present, there are no data suggesting that Marvel D3 affects Leak Pathway permeability, although this has not been exhaustively investigated. In contrast, and as described above, both occludin and tricellulin appear to have roles in regulating Leak Pathway permeability. In most studies, knockdown of occludin does not alter Leak Pathway permeability [79,128,129,130,131,132,134]. This may reflect the fact that occludin knockdown or knockout induces a redistribution of tricellulin from the typical tricellular junction localization to bicellular junctions [79,200]. In MDCK II cells, which express low levels of tricellulin, tricellulin knockout did not alter Leak Pathway permeability [79], whereas, in HT-29/B6 cells, which express higher tricellulin levels, tricellulin knockdown increased permeability to 4 kDa and 10 kDa fluorescein-dextrans [111]. Knockout of both occludin and tricellulin in MDCK II cells increased permeability to both 4 kDa and 10 kDa fluorescein-dextrans, although it was unclear if permeability was increased for larger macromolecules [79]. This was associated with decreased complexity of the tight junction strand organization, as evidenced by fewer horizontal strands and fewer strand branch points in the double knockout cells [79]. Reexpression of either occludin or tricellulin in these double knockout cells restored both the strand complexity and the lower macromolecule permeability.

It is tempting to speculate that, while claudins form the physical barrier of the strand structure, occludin and tricellulin regulate the complexity of this structure through modulation of claudin strand branching/connections. As described by Tervonen et al. [114], less crosslinking and/or longer lengths of unbranched strands could increase the frequency and, possibly, open time before reannealing for strand breaks which would increase Leak Pathway permeability. This hypothesis requires further investigation.

## 12. Conclusions and Outlook

The epithelial paracellular permeability pathway for macromolecules, the Leak Pathway, has received increasing attention and experimental investigation in recent years. Initial studies suggested the Leak Pathway represented sites of unrestricted paracellular passage, possibly due to cell damage. While this cell damage pathway, the Unrestricted Pathway, is still recognized, studies from many laboratories around the world have demonstrated that the Leak Pathway is a distinct paracellular permeability pathway localized to tight junctions and exhibiting dynamic regulation. Studies have described the regulation of Leak Pathway permeability by a wide variety of conditions and compounds. Responses are, in some cases, cell type-specific, arguing for caution about generalizing responses across epithelia. Tension/Stress on the tight junction structure may provide a unifying mechanism underlying various regulatory events in different epithelia, but this requires further study. Multiple tight junction proteins and cellular proteins have been identified that can influence Leak Pathway permeability under various conditions and in different epithelial cell types.

Although much information about the Leak Pathway has been reported, many fundamental questions remain. For example, it is still unclear whether the Leak Pathway opening is comprised of a specific macromolecular complex of tight junction and cellular proteins, similar to the claudin pores or the gap junction, or is simply a disrupted region of the tight junction structure. Perhaps, Leak Pathway openings represent weak points in the overall tight junction structure with a distinct composition/organization that make them more easily disrupted by imposed cellular or external tension. Other fundamental questions relate to the size of the Leak Pathway opening and its dynamic behavior. While some studies have reported a size limit for the Leak Pathway of ~60 Å, this does not explain the ability of larger macromolecules to cross the epithelium. It is likewise unclear whether the Leak Pathway represents a progressive movement of macromolecules through the tight junction structure or the movement of macromolecules through a disruption of the entire tight junction structure or both modes of permeation. While these questions have intrinsic value for acquiring a better understanding of the Leak Pathway, they are also highly relevant for understanding the role of the Leak Pathway in the normal physiology and pathophysiology of various epithelial tissues. In addition, developing strategies to deliver macromolecules across epithelial barriers has been a major interest of drug companies. That interest has only increased with development of macromolecule pharmaceuticals (biologics) that currently are delivered through infusion. A greater understanding of the Leak Pathway may enable the development of novel approaches for macromolecule drug delivery.

To date, most studies of the Leak Pathway have examined the bulk transepithelial movement of macromolecules at the cell population level. While these approaches still have utility, the questions raised above likely will also require new analytical approaches that yield information at a more molecular level. Imaging methodologies, such as super-resolution microscopy and single molecule tracking, may be required to define the properties and dynamics of individual Leak Pathway openings. Identification of the Leak Pathway opening structure and its constituents may require novel approaches to visualize the sites of transepithelial macromolecule passage and to identify the individual components at these sites. The use of tight junction proteins modified to sense and report changes in tension, such as the ZO-1 tension sensor construct [197], may enable analysis of the role of tension on specific tight junction proteins and the tight junction structure as a whole under normal and stress conditions. A potential complication with using these types of approaches may be the potentially limited number of Leak Pathway openings within an epithelial cell population and the dynamic behavior of these openings. We look forward with anticipation to the application of new approaches to analysis of the Leak Pathway. They will likely yield significant progress and, also, some surprises in the evolution of our understanding of this paracellular epithelial permeability pathway.

## Figures and Tables

**Figure 1 ijms-22-07677-f001:**
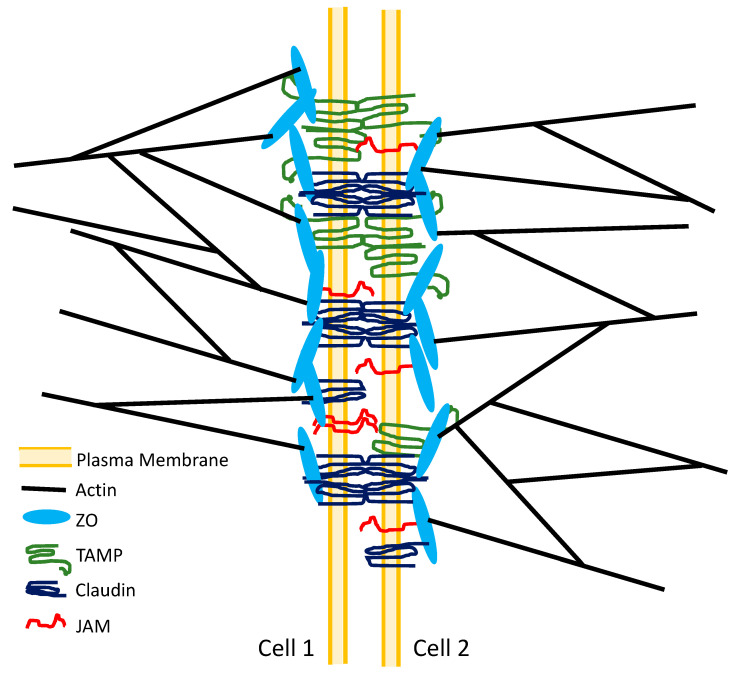
Side view of a model of tight junction supramolecular organization. The tight junction membrane proteins form homo- and heterodimers within the same membrane and between apposing membrane faces [28,32,33,34,35,36,37]. The tight junction membrane proteins are crosslinked to the underlying actin cytoskeleton and microtubular network (not shown) by ZO proteins [16,19,38,39,40,41,42,43,44,45,46] and cingulin, respectively. The claudin proteins organize into an anastomosing network of antiparallel double strands [28,32,33]. Occludin proteins are localized to bicellular junctions and exhibit a modest preferential localization at claudin strand branching sites [62]. See the text for a more complete description of the proposed tight junction organization. The possible involvement of lipids in the tight junction structure is not shown in this figure.

**Figure 2 ijms-22-07677-f002:**
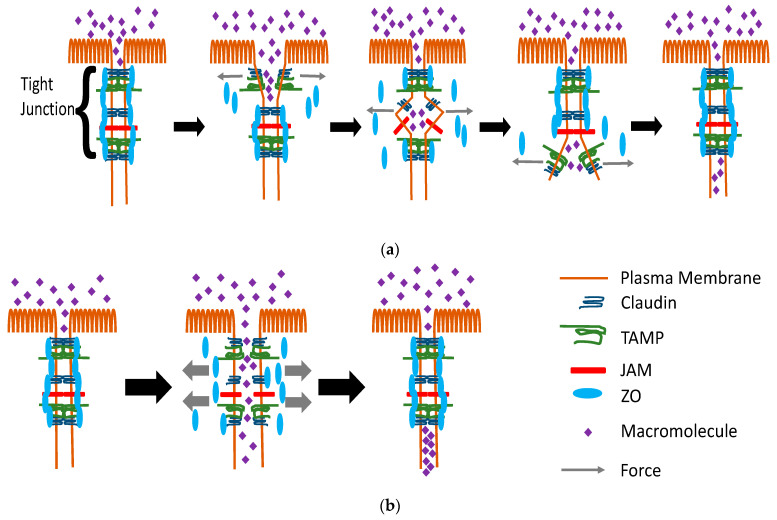
Illustrations of alternative mechanisms of Leak Pathway permeability. (**a**) Breaks in individual tight junction strands enable the progressive movement of macromolecules from a donor fluid compartment to a receiver fluid compartment, passing from one intra-tight junction compartment to the next via the strand breaks. This leads to bursts of macromolecules entering the receiver fluid compartment. (**b**) The entire tight junction structure is disrupted at a specific locus leading to formation of a continuous opening joining the donor and receiver fluid compartments. Macromolecules move through this opening from one compartment to the other in a steady stream until the opening is resealed.

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
