# Peer review of "The Epithelial Cell Leak Pathway"

_ijms, 2021, doi:10.3390/ijms22147677_

Round 1

Reviewer 1 Report

This is a comprehensive review of tight junctions particularly the leak Pathway. Even though the molecular mechanism, its properties, and its regulation of the Leak Pathway remain unclear. The authors provided a detailed historical background to the evolution of the leak Pathway concept and current theories for the Leak Pathway. The review was well organized, well written, and easy to understand.

One minor comment, please add a space between “left” and “hand” in line 163.

Author Response

“One minor comment, please add a space between ‘left’ and ‘hand’ in line 163.” – The sentence has been deleted.

Reviewer 2 Report

The submitted review of Monaco and colleagues deals with the topic of the different paracellular pathways, in particular the leak pathway but giving an insight into the pore pathway in comparison as well.
Although being of interest, there are already several reviews published dealing with this topic. Thus, it might be worth to give not only an up-to-date review, but maybe focus on new aspects to significantly differ from the already present literature about the leak pathway. Several points that might be taken in account are listed below.

Another thing that should be taken in account is that there is already a manuscript from the same authors available as pre-print that has exactly the same title but differs in content. It would be good to use another title to differentiate from this other article.

Major points:

  1. Several aspects might be good to be mentioned. For example, the non-restrictive pathway (though more relevant under pathological conditions) is not mentioned at all. It is only vaguely described under the section dealing with the „Cell Damage/Death Model”.

In addition, it might be worth to mention that in endothelia, the macromolecule passage might be different, e.g. here the tricellular route seems to be dominant (Ghim M, Mohamied Y, Weinberg PD. The Role of Tricellular Junctions in the Transport of Macromolecules Across Endothelium. Cardiovasc Eng Technol. 2021 Feb;12(1):101-113. doi: 10.1007/s13239-020-00483-x).

  1. In line 58 and 59 the authors should not only mention that there is categorization into “tight” and “leaky”, but also should shortly explain it, especially as this is still used and of importance.
  2. Figure 1, right image. The image appears a little bit too speculative and does not give supportive information for the description of pore and leak pathways. What is known about the localization of JAM-A? Why does it appear to swim around somewhere in the area of the TJ? As JAM-A is required for the stable apical junction localization of ZO-1, would one not expect it to be as focused as ZO-1 and membrane TJ proteins? If only occludin is shown, why is the legend saying the green spots are TAMP? What is known about the localization of MARVEL D3 and tricellulin in the TJ network?
  3. There are some doublings of information about claudins that might be shortened or removed in one or the other section, especially when the focus of this review is more on the leak than the pore pathway.
  4. The observed concentration of occludin at strand branches suggests a role here, however there are still many occludin molecules localized at the normal strands (see [65] or Fujimoto, Journal of Cell Science 108, 3443-3449 (1995), https://doi.org/10.1242/jcs.108.11.3443). Thus, one should be careful in underlining this observation as a clear preference. Nevertheless, there are new findings that point also in the direction of an important role for occludin (and also tricellulin) in organizing the branching and complexity [82], also as mentioned. In this context, it also might be interesting to add and shortly discuss the complex interaction and replacing each other roles of occludin and tricellulin as this also appears to have significant influence on macromolecule permeability and thus the leak pathway, even if the location of bi- or tricellular for this might not be solved up to date. Maybe it would be worth to speculate about mixed contributions, which maybe also may differ depending on the epithelial background (e.g. tight and leaky, also see under 8.).
  5. The ideas for the progressive movement in the strand break model are not new. Please add the original citations here, for example Anderson JM, Van Itallie CM, Fanning AS. Setting up a selective barrier at the apical junction complex. Curr Opin Cell Biol. 2004 Apr;16(2):140-5. doi: 10.1016/j.ceb.2004.01.005 and [17].
  6. The authors mention own calculations, but neither show them nor cite the publication(s) where they have been made (line 740). Furthermore, the unpublished data about the “twisting” (line 457-460) should either be explained and shown in more detail or be left out. It is not a good scientific style to mention data no one knows, especially in a review.
  7. The order of subchapters might be in parts rearranged to keep the flow, which could be first introducing the paracellular barrier in general and the TJ, then the two pathways in detail, starting with the pore pathway. Then one should introduce the leak pathway in general, the components and in the regulations for the leak pathway. Then the comparison and connection of both pathways makes sense. The article ends quite abrupt, some concluding remarks and an outlook should be added. Here also the contribution of all models for the leak pathway could be mentioned, e.g. the calculation of [193] underline that the proposed alternatives for the leak pathway, the bicellular strand opening dynamics and the tricellular pores contribute together to different extends, depending on the epithelial properties.

Potential suggestion for and order (and then of course, the introducing first sentences or respective last sentences should be adjusted):

  1. THE TIGHT JUNCTION IS THE PARACELLULAR PERMEABILITY BARRIER
  2. STRANDS, PORES, AND PARACELLULAR PERMEABILITY
  3. TIGHT JUNCTION STRUCTURE
  4. WHAT CONSTITUTES THE PARACELLULAR PERMEABILITY BARRIER?
  5. TWO PARACELLULAR PERMEABILITY PATHWAYS: THE PORE PATHWAY AND THE LEAK PATHWAY
  6. WHAT IS THE LEAK PATHWAY?
  7. WHAT CELL COMPONENTS ARE PART OF THE LEAK PATHWAY?
  8. DOES TIGHT JUNCTION STRESS/TENSION AFFECT LEAK PATHWAY PERMEABILITY?
  9. HOW CAN LEAK PATHWAY PERMEABILITY BE REGULATED INDEPENDENTLY FROM PORE PATHWAY PERMEABILITY?
  10. PORE PATHWAY VERSUS LEAK PATHWAY: THE SAME OR DIFFERENT?
  11. Conclusion and Outlook

Minor:

  1. Use consistent writing of “claudin-X” or “claudin X”.
  2. After first introduction of MDCK Type II renal epithelial cells, one can say MDCK II. Same accounts for MDCK C7 and MDCK I. On the other hand, one should be correct in the naming of other cell lines as well, e.g. HT29 is not the same as HT-29/B6.
  3. Adherens junction does not have to be written in italics, but zonula adherens.
  4. Citation [162] is the same as [159].
  5. If kept, figure 1 should be divided into A and B. Citations should be referenced in the figure legends, too, for example in Fig. 2 and the statements of Fig. legend 1.
  6. What makes the findings in Figure 2 so special that they are added as image to the review but no other findings?

Author Response

The title is changed. It now explicitly states that this review focuses only on the epithelial cell Leak Pathway. This also addresses the comment regarding mentioning the endothelial cell Leak Pathway. The Leak Pathways in epithelial vs endothelial cells appear to exhibit substantial differences. Documenting and discussing the extent of these differences is beyond the scope of this review.

Major Points

  1. We have included a more extensive description of the Unrestricted Pathway.
  2. We have provided a brief description of the designation of leaky vs tight epithelia.
  3. As indicated by reviewer #2, the specific location of proteins within the tight junction structure is currently speculative. We did not intend to imply a specific localization for proteins within this structure by the righthand panel of Figure 1. We have, therefore, removed the righthand panel of Figure 1.
  4. We have attempted to consolidate the discussion of the claudins.
  5. We have added a section discussing the potential roles of occludin and tricellulin in modulating the organization and complexity of the tight junction structure and how this could affect Leak Pathway permeability.
  6. We have included the indicated references regarding progressive solute movement in the strand break model. We had not intended to omit key initial descriptions of this model and apologize for the omission.
  7. We have provided the calculation of the Stokes-Einstein equation using the tricellular pore dimensions. We have omitted the reference to the cell twisting. This observation is currently under further investigation.
  8. The review sections have been reordered mostly according to the reviewer #2 suggestions. A “Conclusions and Outlook” section has been added at the end of the text. Reference numbers have been corrected accordingly.

We have added a small section on tricellulin in the “What Cell Components Are Part of the Leak Pathway” section.

We have added a Table 1 with data demonstrating the difference in Leak Pathway permeability of different-sized solutes in MDCK I versus MDCK II cells. This is related to the published discussion of the tight junction structure in each cell line and the potential impact of strand length and complexity on Leak Pathway permeability.

Minor Points

  1. The designation of specific claudins is now consistent throughout the document as “claudin-X”.
  2. The cell line names have been corrected as requested.
  3. Adherens junction italics have been removed.
  4. Citations have been corrected throughout the manuscript to address the substantial changes to the text.
  5. The righthand panel of Figure 1 has been deleted. We agree that it was speculative. It was not intended to indicate a specific localization of JAM-A or other TAMPs.
  6. Figure 2 has been removed. It was included to confirm that the ZO-1 knockdown and ZO-2 knockdown had different effects on the Leak Pathway flux but, as noted in the text, this is already published.

Round 2

Reviewer 2 Report

The revised review of Monaco and colleagues has improved a lot and all raised points were taken in account. However, there a still a few minor points they should change to make the improvement sufficient for acceptance.

  1. The authors added a definition for leaky and tight epithelia that needs an addition. Leaky epithelia mostly have high small ion permeability and thus low transepithelial resistance, but an important point is also the ratio of paracellular to transcellular permeability. In leaky epithelia, the permeability of the paracellular route is higher than the transcellular permeability. In tight epithelia, the overall permeability is often low (thus, the high transepithelial resistance) and the paracellular permeability is lower than the transcellular one. However, the citations that are basis of these definitions are correct.
  2. The new table (Table 1) appears to be own data. Please add the methods, maybe as supplement for that if these data cannot be found somewhere else to be cited. In this context, it is also a little bit surprising to have such big differences in the n-number. Is there a reason that at least three times more measurements were done for the MDCK I cells? Are the errors SD or SEM? Usually, one should not put unpublished/unreviewed experiments in a review article. If not essential for understanding one rather should leave that out.

Author Response

Dear Ms. Guo and Reviewer 2:

Thank you for your additional comments regarding our submitted manuscript. We have made the following changes to address those concerns.

  1. The authors added a definition for leaky and tight epithelia that needs an addition. Leaky epithelia mostly have high small ion permeability and thus low transepithelial resistance, but an important point is also the ratio of paracellular to transcellular permeability. In leaky epithelia, the permeability of the paracellular route is higher than the transcellular permeability. In tight epithelia, the overall permeability is often low (thus, the high transepithelial resistance) and the paracellular permeability is lower than the transcellular one. However, the citations that are basis of these definitions are correct.

We apologize for the shorthand description of “leaky” versus “tight” epithelia we used in the previous version of the manuscript. As Reviewer 2 correctly pointed out, there is more than just high versus low small ion permeability to this definition. We have attempted to provide a more complete description of the distinction while also striving for brevity since this is not the major focus of the manuscript.

  1. The new table (Table 1) appears to be own data. Please add the methods, maybe as supplement for that if these data cannot be found somewhere else to be cited. In this context, it is also a little bit surprising to have such big differences in the n-number. Is there a reason that at least three times more measurements were done for the MDCK I cells? Are the errors SD or SEM? Usually, one should not put unpublished/unreviewed experiments in a review article. If not essential for understanding one rather should leave that out.

We had included Table 1 containing new data to provide some limited support for the hypothesis that strand complexity is a factor in Leak Pathway permeability. We agree with Reviewer 2 that it is not appropriate to provide new data without a more complete and thorough presentation of methods and analysis. Since we do not believe the data are essential, we have removed Table 1 and omitted the mention of these data.

Thank you again for your consideration and input of our submitted manuscript. We look forward to hearing from you.

Regards,

Kurt Amsler